# Factors related to a sense of economic insecurity among older adults who participate in social activities

**Yuriko Inoue**[1]*, **Hisae Nakatani**[1], **Ichie Ono**[2], **Xuxin Peng**[1]

1 Graduate School of Biomedical and Health Sciences, Hiroshima University, Hiroshima City, Hiroshima, Japan, 2 Department of Nursing, Yasuda Women's University, Hiroshima City, Hiroshima, Japan

* hollyhocklily@gmail.com

## Abstract

Japan has the highest life expectancy worldwide. Older adults who experience economic insecurity may refrain from seeking medical consultation or using long-term care insurance, and these behaviors may increase the incidence and progression of frailty. This study conducted a cross-sectional survey to identify factors related to a sense of economic insecurity among older adults who participate in social activities, and identified support measures. In total, 1,351 older adults aged ≥65 years who had participated in social activities voluntarily completed an anonymous self-administered questionnaire. The questionnaire encompassed their physical, cognitive, social, and psychological conditions, and economic insecurity. We performed univariate analysis considering a sense of economic insecurity as the dependent variable, and conducted multiple logistic regression analysis (forced entry method) considering the independent variables with p<0.1 as the covariates. Among the 872 filled questionnaires, 717 were analyzed as they had no missing data with respect to the responses to survey questions (valid response rate was 53.1%). Analysis results showed that 43.6% of the older adults had a sense of economic insecurity, which was most common among those aged 75–84 years, accounting for 47.3%, followed by those aged 65–74 years accounting for 44.1%, and those aged ≥85 years accounting for 31.5% (p<0.05). The sense of economic insecurity was not associated with physical conditions, subjective symptoms of dementia, or social conditions; however, it grew with increased loneliness (OR: 1.71, 1.002–2.92, p = 0.049) and decreased with an increased subjective sense of well-being (OR: 0.86, 0.81–0.92, <0.001). Economic insecurity among older adults was not associated with physical, cognitive, or social aspects, as reported in previous studies. The survey respondents constituted older adults who participate in social activities. Maintaining interactions within the community, even in old age, may prevent loneliness and improve subjective health.

## Introduction

Japan has the highest life expectancy worldwide, reaching 81.5 years for men and 87.6 years for women after 2020 [1]. The population of older adults aged ≥85 years continues to increase,

**Data Availability Statement:** All relevant data are within the paper and its Supporting Information files.

**Funding:** IY FBK220531026 France Bed Medical Home Care Research Subsidy Public Incorporated

Foundation https://www.fbm-zaidan.or.jp/subsidy/index.html The funders had no role in study design, data collection and analysis, decision to publish, or preparation of the manuscript. IY JPMJFS2129 Hiroshima University Research Fellowship https://fellowship.hiroshima-u.ac.jp/research/ The funders had no role in study design, data collection and analysis, decision to publish, or preparation of the manuscript.

**Competing interests:** The authors have declared that no competing interests exist.

and Japan is about to enter the era of the 100-year life ahead of the rest of the world. However, the healthy life expectancy, a period with no restrictions in daily life, and average life expectancy differ by 10.2 years [2]. Living a healthy old age life without needing nursing care, even in longevity, has become an issue. Frailty is a state of increased vulnerability to changes in health status resulting from an age-related decline in physiological function [3]. Notably, the concept of frailty has been expanded beyond its physical aspects to include psychological and social aspects [4]. As the number of frail older adults increases with aging [5], preventive activities to maintain physical and mental function are important. Feng et al. reported that the risk factors associated with incident or increased frailty include low income among older adults [6]. According to a report by the Ministry of Health, Labour and Welfare, older adults account for a high percentage of the suicide mortality rate in Japan [7], and individuals aged ≥60 years account for 37.7% of the total number of suicide cases. In 2022, the most common causes and motives of suicide among older adults in Japan were "health problems," followed by "family problems" and "economic and lifestyle problems" [8]. In conclusion, these trends suggest that there may be an association between a sense of economic insecurity among older adults and their physical and mental health. De Leo reported that suicide prevention for older adults should focus on the number of socio-environmental conditions that can be particularly worrisome in old age, such as decreased physical health, social isolation and loneliness, and economic insecurity [9]. Haines et al. reported that feelings of financial dissatisfaction adversely affect an individual's subjective health [10]. According to a survey by the Cabinet Office of Japan [11], 59.0% of individuals aged 60–69 years and 35.8% of individuals aged ≥70 years were worried and insecure about their future economic prospects; therefore, older adults with economic insecurity may refrain from using medical care and long-term care insurance. This may lead to the incidence and progression of frailty.

In Japan, the Ministry of Health, Labour and Welfare (MHLW) encourages older adults to engage in social interactions with their neighbors and participate in social activities to prevent them from needing nursing care and becoming frail [12,13]. Examples of such activities include exercise and hobby activities in venues planned by health professionals in which older adults, regardless of their health status, can voluntarily participate. Participation in social activities costs nothing to approximately 200 yen (0 to approximately 1 US dollar) per month, and may decrease the incidence of functional disability [14] and cognitive decline [15]. A previous study analyzing differences in the prevalence of and factors associated with frailty in five Japanese residential areas reported that the higher level of social activities is attributable to the lower prevalence of frailty, and in most areas, a subjective economic status were significantly associated with frailty [16]. Encouraging participation in social activities, especially high-frequency social participation, could decrease the risk of and reverse the progression of frailty among middle-aged and older populations [17]. For low-cost social activities to function as a place for preventing frailty and suicide, it is necessary to investigate the actual state of the participants' economic insecurity and discuss how to provide support for effective social activities. However, few historical studies specifically focus on older adults participating in social activities. Therefore, this study conducted a cross-sectional survey to identify factors related to a sense of economic insecurity among older adults who participate in social activities (not limited to older adults experiencing frailty) and to discuss support measures.

## Materials and methods

### Research design

An anonymous self-administered questionnaire was used in this cross-sectional study.

## Research participants

The participants of the study were older adults aged 65 years or older who voluntarily partici-pated in low-cost community-based social activities, such as exercise and hobby activities, and who were able to fill out a self-administered survey form. Based on statistical power calcula-tions (power: 0.80, alpha value: 0.05), the sample size was 602 [18,19]. The collection rate was assumed to be around 45% in referring to previous studies [20]. We asked four community general support centers in Hiroshima City, Japan that had agreed to cooperate in this study to distribute the survey instrument. The questionnaire was distributed to 1351 older adults who participated in social activities during the survey period.

## Survey methods

For the survey, the staff of the community general support centers explained in writing and orally the significance, purpose, methods, and ethical considerations of this study. These aspects are stated in the survey request letter, and the questionnaire was distributed among older adults who participate in social activities by hand. Individual older adults who received the questionnaires voluntarily completed it and then returned it either by posting in a sealed envelope or bringing it back to the collection box set up at the facilities where social activities were held. The survey was conducted from July to December 2022.

## Survey contents

**1) Basic attributes.** Basic attributes constituted gender, age, household composition, mar-ital status, spouse status, and whether the participant had a child (children). Respondents who selected "married" in response to marital status were also asked to state their spouse status (cohabiting, bereaved, divorced, or separated) in a multiple answer question. For the financial characteristic, we considered asking participants about their current income/savings and expenditures; however, we anticipated low response rates some older adults may be reluctant to answer questions relating to their personal information or may find it difficult to answer these questions. We also felt that using a large number of scales may be physically and mentally taxing for older adults. Therefore, we asked the participants about their subjective sense of anxiety about their economic situation. To determine their sense of economic insecurity, we asked the participants, "Do you have economic anxieties?" and asked them to respond using a four-point Likert scale (not worried, not too worried, a little worried, worried).

**2) Physical conditions and subjective symptoms of dementia.** Regarding the physical conditions and subjective symptoms of dementia, we inquired about the presence of illness, whether long-term care insurance services were used, and subjective symptoms of dementia. Subjective symptoms of dementia were assessed using a self-administered dementia checklist [21–23]. This checklist was developed to enable community-dwelling older adults to identify the decline in their cognitive and daily living functioning by filling out a self-administered form. The ten question items, including five items for subjective cognitive decline observed in the early stages of dementia and five items for subjective daily living functioning, were answered based on a four-point Likert scale for cognitive and daily living functioning, and sig-nificant correlations with the Mini-Mental State Examination (MMSE) [24] and Clinical Dementia Rating (CDR) [25] were confirmed. The score ranges from 10–40, with higher scores indicating greater severity of subjective dementia symptoms.

**3) Social conditions.** Regarding social conditions, we asked about the frequency of out-ings, participation in social activities, working status, and the presence or absence of social iso-lation. The frequency of outdoor activities and participation in social activities was based on a six-point Likert scale (daily, 5–6 times a week, 3–4 times a week, 1–2 times a week, 1–3 times a

month, and less than once a month). Social isolation was assessed using the Lubben Social Network Scale (LSNS-6) [26]. The LSNS-6 is used to determine the number of people in a network using a six-point Likert scale (none, one, two, three or four, five through eight, nine or more) to determine emotional and instrumental support from family and relatives (three items) and friends, including those in the neighborhood (three items). The score ranges from 0–30; the more people there are in the network, the higher the score. A score below 12 points indicates social isolation.

**4) Psychological conditions.** Regarding psychological conditions, we assessed loneliness and subjective well-being. Loneliness was assessed using the University of California, Los Angeles Loneliness Scale, version 3. The UCLA Loneliness Scale (Loneliness) measures loneliness from the situational position with 20 items [27] or three items (short version) [28]. In this study, a shortened three-item version was used to improve the conciseness of the participants' responses. The score ranges from 3–9; the higher the score, the lonelier the participant is, and a score of $\geq 6$ indicates loneliness. Subjective wellbeing was assessed using the Philadelphia Geriatric Center Morale Scale (11-item) [29,30] developed for older adults. Eleven items (short version) were used in this study, and the questions were answered using a three-point Likert scale (yes, no, do not know). A positive "yes" response to subjective well-being is assigned a score of 1, and "no" or "do not know" a score of 0. The scores range from 0 to 11, with higher scores indicating a higher sense of well-being.

## Analysis methods

The validity of the results is questionable if each scale used has missing values of 5% or more [31]. Therefore, the survey questionnaires without missing data on basic attributes, subjective symptoms of dementia, LSNS-6, loneliness, and subjective well-being were analyzed. Given that life expectancy is increasing around the world, with the number of older adults aged 85 and over accordingly growing, analyses of the social participation of older adults must take into account differences in age [32]. Therefore, age was categorized into three groups: 65–74, 75–84, and $\geq 85$ years, based on the $\geq 65$ age groups defined by the Long-Term Care Insurance Law, to which formal services for older adults in Japan are applicable. The frequency of outings and participation in social activities were categorized into two groups: at least once a week more and less than once a week, which are indicators of confinement [33] and criteria for lack of human contact [34]. Subjective symptoms of dementia were divided into "no" for scores below 18 and "yes" for scores of $\geq 18$, based on previous studies [22]. The sense of economic insecurity was categorized into two groups: "not worried" and "not too worried" were classified as "not worried" and "a little worried" and "worried" were classified as "worried." With regard to statistical analysis, we compared the differences between the two groups with respect to economic insecurity and basic attributes, physical, cognitive, and social conditions using the $\chi^2$ test, and subjective well-being differences using the Mann-Whitney U-test. For multiple comparisons among the three age groups and basic attributes (physical, cognitive, and social), we used z-tests (the Bonferroni method for adjusting p-values), and subjective well-being differences using the Kruskal-Wallis test. To elucidate the factors related to economic insecurity for the "worried" group, we performed a univariate analysis considering the sense of economic insecurity as the dependent variable, and a performed multiple logistic regression analysis (forced entry method) with the independent variables as the covariates. The significance level of the covariates to be input into the model was set to less than 0.1. In the analysis, we set dummy variables, where 0 was assigned for "not worried" and 1 for "worried" with respect to sense of economic insecurity. For gender, we set 0 for male and 1 for female. For age, we set 0 for the 65–74 age group, 1 for the 75–84 group, and 2 for $\geq 85$ age group. For household

composition, we set 0 for "with spouse/children/others" and 1 for "living alone." For marital status, we set 0 for "married" and 1 for "unmarried." For spouse status, we set 0 for "no" and 1 for "yes" for each item of bereaved, divorced, and separated. For working status, we set 0 for "yes" and 1 for "no." For subjective symptoms of dementia, we set 0 for "no" and 1 for "yes." For social isolation, we set 0 for "no" and 1 for "yes." For loneliness, we set 0 for "not lonely" and 1 for "lonely." For subjective well-being, we used the total score of 0–11. Data were analyzed using the statistical software SPSS Version 27. A two-tailed p-value of <0.05 was considered to indicate statistical significance.

### Ethical considerations

In terms of ethical considerations, the research participants received written and oral communication that participation was voluntary, that there would be no disadvantage from declining to participate, and that there would be no description of information that might lead to the identification of personal information. The request letter clearly stated that turning in the filled questionnaire would be regarded as consent, and that the questionnaire could not be withdrawn after it had been posted or dropped into the collection box. Participants were requested to indicate their agreement to participate by ticking a box, and their consent was obtained in writing. This study was approved by the Ethics Committee for Epidemiological Research of Hiroshima University (approval number: E2022-0020; approval date: June 9, 2022).

## Results

Of the 872 filled questionnaires (response rate: 64.5%), 155 were excluded owing to missing data, and 717 were analyzed (valid response rate: 53.1%) (Fig 1, S1 Table).

The scales used in the study were tested for normality (Shapiro-Wilk test) and did not follow a normal distribution (P<0.01). The reliability coefficients (Cronbach's alpha coefficients) for each scale were as follows: self-administered dementia checklist: 0.822; LSNS-6: 0.863; loneliness: 0.810; and subjective well-being: 0.758.

### Characteristics based on age group

Table 1 shows the participant characteristics according to age group. Of these, 17.4% were male and 82.6% were female. The mean age was 78.7±6.0 years (65–96 years). In terms of age group, 25.9% were 65–74 years old, 56.3% were 75–84 years old, and 17.7% were ≥85 years old. Compared to the two age groups: 65–74 and 75–84 age groups, there were more males in the ≥85 age group (27.6%; p<0.05). In terms of household composition, 66.9% lived with their family and 33.1% lived alone. Most older adults living alone were identified in the ≥85 age group (50.4%; p<0.05). In total, 97.6% were married and 42.3% of the older adults were bereaved of their spouses. With regard to the sense of economic insecurity, 43.6% felt insecure. The percentage of older adults with a sense of economic insecurity was highest in the 75–84 age group at 47.3%, followed by the 65–74 age group at 44.1% and ≥85 age group at 31.5%; differences were observed among age groups (p<0.05).

Regarding physical conditions and subjective symptoms of dementia, 85.4% of the older adults suffered from certain illness, and there were more older adults attending the hospital in the 75–84 age group which accounted for 90.1% and the ≥85 age group accounting for 89.0% than in the 65–74 age group at 72.6% (p<0.05). In addition, 9.5% of the older adults had subjective symptoms of dementia, and the percentage increased with age, with 2.2% in the 65–74 age group, 8.2% in the 75–84 age group and 24.4% in the ≥85 age group (p<0.05). Moreover, 10.7% of the older adults used long-term care insurance. Older adults going out less than once a week accounted for 7.5, 8.7, and 22.0% in the 65–74, 75–84, and ≥85 years groups (p<0.05).

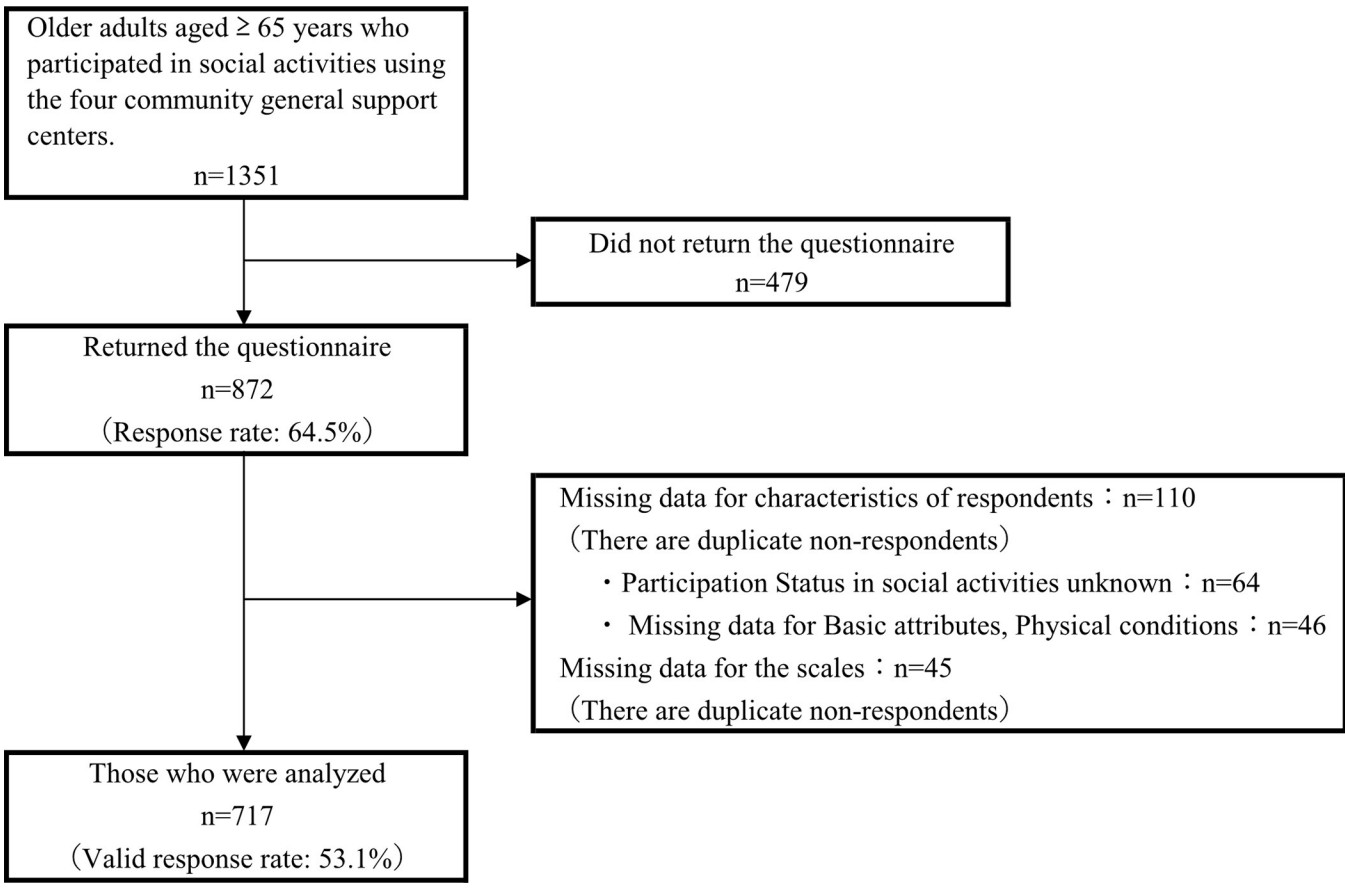

**Fig 1. Flowchart of the study participants.**

No differences in frequency of participation in social activities were observed among the age groups. In the 65–74 years group, 18.3% accounted for working older adults (p<0.05). No difference was observed in social isolation among age groups, as well as in the psychological conditions of loneliness and subjective well-being.

### Factors related to sense of economic insecurity

Table 2 shows the associations of basic attributes, physical conditions, subjective symptoms of dementia, social conditions, and psychological conditions with a sense of economic insecurity. No significant differences were observed in sense of economic insecurity by gender, household composition, marital status, or having a child (children). Among the physical conditions and subjective symptoms of dementia, no difference was found between older adults with economic insecurity and older adults without economic insecurity in all of the items of illness, subjective symptoms of dementia, and the use of long-term care insurance. Among the social conditions, there was no difference between older adults feeling economically insecure and older adults not feeling economically insecure in terms of frequency of outings and participation in social activities, working status, and social isolation. Among psychological conditions, older adults experiencing economic insecurity had higher rates of loneliness (< .001) and lower subjective well-being (< .001).

Table 3 shows the results of the multiple logistic regression analysis. The five covariates in univariate analysis (p<0.1: age, marital status, working status, loneliness, and subjective well-

**Table 1. Characteristics according to age group.**

| | All | | 65–74 | | 75–84 | | 85+ | | p-value | Effect size |
|---|---|---|---|---|---|---|---|---|---|---|
| | (n = 717) | | (n = 186) | | (n = 404) | | (n = 127) | | | |
| **Basic attributes** | | | | | | | | | | |
| Gender | | | | | | | | | | |
| Male | 125 | (17.4) | 27 | (14.5) | 63 | (15.6) | 35 | (27.6) [ab] | 0.004 [†] | 0.124 |
| Female | 592 | (82.6) | 159 | (85.5) | 341 | (84.4) | 92 | (72.4) [ab] | | |
| Household composition | | | | | | | | | | |
| With spouse/children/others | 480 | (66.9) | 141 | (75.8) | 276 | (68.3) | 63 | (49.6) [ab] | <0.001 [†] | 0.184 |
| Living alone | 237 | (33.1) | 45 | (24.2) | 128 | (31.7) | 64 | (50.4) [ab] | | |
| Marital Status (n = 714) | | | | | | | | | | |
| Married | 700 | (98.0) | 178 | (95.7) | 395 | (98.5) | 127 | (100.0) | | |
| Unmarried | 14 | (2.0) | 8 | (4.3) | 6 | (1.5) | 0 | (0.0) | | |
| Spouse Status (n = 700) | | | | | | | | | | |
| Cohabiting | | | | | | | | | | |
| No | 344 | (49.1) | 53 | (29.8) | 198 | (50.1) [a] | 93 | (73.2) [ab] | <0.001 [†] | 0.284 |
| Yes | 356 | (50.9) | 125 | (70.2) | 197 | (49.9) [a] | 34 | (26.8) [ab] | | |
| Bereaved | | | | | | | | | | |
| No | 404 | (57.7) | 140 | (78.7) | 222 | (56.2) [a] | 42 | (33.1) [ab] | <0.001 [†] | 0.302 |
| Yes | 296 | (42.3) | 38 | (21.3) | 173 | (43.8) [a] | 85 | (66.9) [ab] | | |
| Divorced | | | | | | | | | | |
| No | 667 | (95.3) | 165 | (92.7) | 379 | (95.9) | 123 | (96.9) | 0.154 [†] | 0.073 |
| Yes | 33 | (4.7) | 13 | (7.3) | 16 | (4.1) | 4 | (3.1) | | |
| Separated | | | | | | | | | | |
| No | 685 | (97.9) | 176 | (98.9) | 385 | (97.5) | 124 | (97.6) | 0.550 [†] | 0.041 |
| Yes | 15 | (2.1) | 2 | (1.1) | 10 | (2.5) | 3 | (2.4) | | |
| Having a child (Children) | | | | | | | | | | |
| Yes | 670 | (93.4) | 167 | (89.8) | 381 | (94.3) | 122 | (96.1) | 0.050 [†] | 0.091 |
| No | 47 | (6.6) | 19 | (10.2) | 23 | (5.7) | 5 | (3.9) | | |
| Sense of economic insecurity | | | | | | | | | | |
| Not worried | 404 | (56.3) | 104 | (55.9) | 213 | (52.7) | 87 | (68.5) [b] | 0.007 [†] | 0.117 |
| Worried | 313 | (43.7) | 82 | (44.1) | 191 | (47.3) | 40 | (31.5) [b] | | |
| **Physical Conditions** | | | | | | | | | | |
| Illness | | | | | | | | | | |
| No | 105 | (14.6) | 51 | (27.4) | 40 | (9.9) [a] | 14 | (11.0) [a] | <0.001 [†] | 0.214 |
| Yes | 612 | (85.4) | 135 | (72.6) | 364 | (90.1) [a] | 113 | (89.0) [a] | | |
| Subjective symptoms of dementia | | | | | | | | | | |
| No | 649 | (90.5) | 182 | (97.8) | 371 | (91.8) [a] | 96 | (75.6) [ab] | <0.001 [†] | 0.252 |
| Yes | 68 | (9.5) | 4 | (2.2) | 33 | (8.2) [a] | 31 | (24.4) [ab] | | |
| Using long-term care insurance services | | | | | | | | | | |
| No | 640 | (89.3) | 181 | (97.3) | 381 | (94.3) | 78 | (61.4) [ab] | <0.001 [†] | 0.419 |
| Yes | 77 | (10.7) | 5 | (2.7) | 23 | (5.7) | 49 | (38.6) [ab] | | |
| **Social Conditions** | | | | | | | | | | |
| Frequency of outing | | | | | | | | | | |
| Once a week+ | 640 | (89.3) | 172 | (92.5) | 369 | (91.3) | 99 | (78.0) [ab] | <0.001 [†] | 0.170 |
| Less than once a week | 77 | (10.7) | 14 | (7.5) | 35 | (8.7) | 28 | (22.0) [ab] | | |
| Frequency of participation in social activities | | | | | | | | | | |
| Once a week+ | 609 | (84.9) | 164 | (88.2) | 342 | (84.7) | 103 | (81.1) | 0.222 [†] | 0.065 |
| Less than once a week | 108 | (15.1) | 22 | (11.8) | 62 | (15.3) | 24 | (18.9) | | |

(*Continued*)

**Table 1.** (Continued)

| | **All** | | **65–74** | | **75–84** | | **85+** | | **p-value** | **Effect size** |
|---|---|---|---|---|---|---|---|---|---|---|
| | **(n = 717)** | | **(n = 186)** | | **(n = 404)** | | **(n = 127)** | | | |
| Working Status (n = 710) | | | | | | | | | | |
| Yes | 84 | (11.8) | 34 | (18.3) | 39 | (9.8) [a] | 11 | (8.8) | 0.006 [†] | 0.119 |
| No | 626 | (88.2) | 152 | (81.7) | 360 | (90.2) [a] | 114 | (91.2) | | |
| Social Isolation | | | | | | | | | | |
| No | 484 | (67.5) | 121 | (65.1) | 280 | (69.3) | 83 | (65.4) | 0.503 [†] | 0.044 |
| Yes | 233 | (32.5) | 65 | (34.9) | 124 | (30.7) | 44 | (34.6) | | |
| **Psychological Conditions** | | | | | | | | | | |
| Loneliness | | | | | | | | | | |
| Not lonely | 637 | (88.8) | 163 | (87.6) | 361 | (89.4) | 113 | (89.0) | 0.825 [†] | 0.023 |
| Lonely | 80 | (11.2) | 23 | (12.4) | 43 | (10.6) | 14 | (11.0) | | |
| Subjective Well-Being | 6.3 | (2.74) | 6.6 | (2.73) | 6.3 | (2.71) | 5.92 | (2.81) | 0.090 [‡] | 0.180 |

[†]: $\chi^2$ and z-tests were performed to compare column proportions. p-values have been Bonferroni corrected

a: vs 65–74 years old

b: vs 75–84 years old, p<0.05.

[‡]: The Philadelphia Geriatric Center Morale Scale, Kruskal-Wallis test.

Marital status and work were tested, excluding those who did not want to answer the question.

Data is presented as either n (%) or mean±standard deviation.

being, were input into Model 1. The odds ratio of older adults with economic insecurity was 0.55 (95% confidence interval: 0.33–0.91, p = 0.019) for age 85 years and older compared to 65–74 years for age, and 0.87 (95% confidence interval: 0.82–0.92, P<0.001) for subjective well-being. In this model, the maximum VIF value was 1.127, which did not exceed 10, indicating no multicollinearity. In Model 2, when adjusted for gender, household status, marital status, and spouse status (bereaved or separated), which may affect income and savings in addition to Model 1, the odds ratios were 0.49 (95% confidence interval: 0.29–0.83, P = 0.008) for age 85 years and older compared to 65–74 years for age, 0.87 (95% confidence interval: 0.82–0.92, P<0.001) for subjective well-being. In this model, the maximum VIF value was 2.235, which did not exceed 10, indicating no multicollinearity. In addition to Model 2, the odds ratios for older adults with economic insecurity in Model 3, adjusted for subjective symptoms of dementia and social isolation to account for physical and cognitive fragility and social ties, were 0.53 (95% confidence interval: 0.31–0.91, P = 0.021) for age 85 years and older compared to 65–74 years for age, 1.71 (95% confidence interval: 1.002–2.92, P = 0.049) for loneliness and 0.86 (95% confidence interval: 0.81–0.92, P<0.001) for subjective well-being. In this model, the maximum VIF value was 2.240, which did not exceed 10, indicating no multicollinearity.

## Discussion

In this study, we examined factors related to a sense of economic insecurity among older adults, and explored support measures.

The older the age group of the participants, the greater the physical and cognitive decline and the greater the need for formal assistance through long-term care insurance were shown in this study. Long-term care insurance data can be used to identify frail individuals at high risk of needing long-term care and provide preventive care and interventions [35]. In Japan, approximately 7–10% of older adults living in the community are frail [36,37]. According to a

**Table 2. Factors associated with sense of economic insecurity.**

| | All | | Sense of economic insecurity | | | | p-value | Effect size |
|---|---|---|---|---|---|---|---|---|
| | | | Not worried | | Worried | | | |
| | (n = 717) | | (n = 404) | | (n = 313) | | | |
| **Basic attributes** | | | | | | | | |
| Gender | | | | | | | | |
| Male | 125 | (17.4) | 78 | (19.3) | 47 | (15.0) | 0.133[a] | 0.056 |
| Female | 592 | (82.6) | 326 | (80.7) | 266 | (85.0) | | |
| Age | | | | | | | | |
| 65–74 | 186 | (25.9) | 104 | (25.7) | 82 | (26.2) | 0.007[a] | 0.117 |
| 75–84 | 404 | (56.3) | 213 | (52.7) | 191 | (61.0) | | |
| 85+ | 127 | (17.7) | 87 | (21.5) | 40 | (12.8) | | |
| Household composition | | | | | | | | |
| With spouse/children/others | 480 | (66.9) | 275 | (68.1) | 205 | (65.5) | 0.467[a] | 0.027 |
| Living alone | 237 | (33.1) | 129 | (31.9) | 108 | (34.5) | | |
| Marital Status (n = 714) | | | | | | | | |
| Married | 700 | (98.0) | 397 | (98.8) | 303 | (97.1) | 0.117[a] | 0.059 |
| Unmarried | 14 | (2.0) | 5 | (1.2) | 9 | (2.9) | | |
| Spouse Status (n = 700) | | | | | | | | |
| Cohabiting | | | | | | | | |
| No | 344 | (49.1) | 187 | (47.1) | 157 | (51.8) | 0.217[a] | 0.047 |
| Yes | 356 | (50.9) | 210 | (52.9) | 146 | (48.2) | | |
| Bereaved | | | | | | | | |
| No | 404 | (57.7) | 232 | (58.4) | 172 | (56.8) | 0.657[a] | 0.017 |
| Yes | 296 | (42.3) | 165 | (41.6) | 131 | (43.2) | | |
| Divorced | | | | | | | | |
| No | 667 | (95.3) | 384 | (96.7) | 283 | (93.4) | 0.040[a] | 0.078 |
| Yes | 33 | (4.7) | 13 | (3.3) | 20 | (6.6) | | |
| Separated | | | | | | | | |
| No | 685 | (97.9) | 388 | (97.7) | 297 | (98.0) | 0.795[a] | 0.010 |
| Yes | 15 | (2.1) | 9 | (2.3) | 6 | (2.0) | | |
| Having a child (Children) | | | | | | | | |
| Yes | 670 | (93.4) | 378 | (93.6) | 292 | (93.3) | 0.883[a] | 0.005 |
| No | 47 | (6.6) | 26 | (6.4) | 21 | (6.7) | | |
| **Physical Conditions** | | | | | | | | |
| Illness | | | | | | | | |
| No | 105 | (14.6) | 65 | (16.1) | 40 | (12.8) | 0.214[a] | 0.046 |
| Yes | 612 | (85.4) | 339 | (83.9) | 273 | (87.2) | | |
| Subjective symptoms of dementia | | | | | | | | |
| No | 649 | (90.5) | 365 | (90.3) | 284 | (90.7) | 0.860[a] | 0.007 |
| Yes | 68 | (9.5) | 39 | (9.7) | 29 | (9.3) | | |
| Using long-term care insurance services | | | | | | | | |
| No | 640 | (89.3) | 358 | (88.6) | 282 | (90.1) | 0.525[a] | 0.024 |
| Yes | 77 | (10.7) | 46 | (11.4) | 31 | (9.9) | | |
| **Social Conditions** | | | | | | | | |
| Frequency of outing | | | | | | | | |
| Once a week+ | 640 | (89.3) | 354 | (87.6) | 286 | (91.4) | 0.108[a] | 0.060 |
| Less than once a week | 77 | (10.7) | 50 | (12.4) | 27 | (8.6) | | |
| Frequency of participation in social activities | | | | | | | | |

*(Continued)*

**Table 2.** (Continued)

| | | All | | Sense of economic insecurity | | | | p-value | Effect size |
|---|---|---|---|---|---|---|---|---|---|
| | | | | Not worried | | Worried | | | |
| | | (n = 717) | | (n = 404) | | (n = 313) | | | |
| Once a week+ | | 609 | (84.9) | 349 | (86.4) | 260 | (83.1) | 0.218[a] | 0.046 |
| Less than once a week | | 108 | (15.1) | 55 | (13.6) | 53 | (16.9) | | |
| Working Status (n = 710) | | | | | | | | | |
| Yes | | 84 | (11.8) | 55 | (13.8) | 29 | (9.3) | 0.068[a] | 0.069 |
| No | | 626 | (88.2) | 344 | (86.2) | 282 | (90.7) | | |
| Social Isolation | | | | | | | | | |
| No | | 484 | (67.5) | 275 | (68.1) | 209 | (66.8) | 0.713[a] | 0.014 |
| Yes | | 233 | (32.5) | 129 | (31.9) | 104 | (33.2) | | |
| **Psychological Conditions** | | | | | | | | | |
| Loneliness | | | | | | | | | |
| Not lonely | | 637 | (88.8) | 375 | (92.8) | 262 | (83.7) | <0.001[a] | 0.144 |
| Lonely | | 80 | (11.2) | 29 | (7.2) | 51 | (16.3) | | |
| Subjective Well-Being | | 6.28 | (2.74) | 6.81 | (2.51) | 5.61 | (2.87) | <0.001[b] | 0.210 |

a: $\chi^2$ test.

b: The Philadelphia Geriatric Center Morale Scale, Mann-Whitney U test.

Marital status and working status were tested, excluding those who did not want to answer the question.

Data is presented as either n (%) or mean±standard deviation.

report on the implementation of the long-term care insurance system in Hiroshima Prefecture, in 2022, the percentage of people certified as having a low long-term care insurance level (support needs: levels 1 and 2, and care needs level 1) in Japan was 9.3% (10.1% in Hiroshima Prefecture), which was similar to the 10.7% rate in this study. Lynch et al. reported that compared to people without difficulties, those with economic difficulties exhibited a 4.6 times decline in subjective cognitive functioning, a 3.38 times decline in independent activities of daily living

**Table 3. Factors related to sense of economic insecurity.**

| | Category | Model 1 | | | Model 2 | | | Model 3 | | |
|---|---|---|---|---|---|---|---|---|---|---|
| | | OR | (95% CI) | p-value | OR | (95% CI) | p-value | OR | (95% CI) | p-value |
| Age | 65–74 (reference) | | | | | | | | | |
| | 75–84 | 1.15 | (0.79–1.66) | 0.473 | 1.08 | (0.74–1.58) | 0.682 | 1.11 | (0.76–0.62) | 0.603 |
| | 85+ | 0.55 | (0.33–0.91) | 0.019 | 0.49 | (0.29–0.83) | 0.008 | 0.53 | (0.31–0.91) | 0.021 |
| Divorced | Yes (reference: No) | 1.85 | (0.88–3.90) | 0.105 | 2.23 | (0.99–5.04) | 0.053 | 2.18 | (0.96–4.92) | 0.061 |
| Working Status | No (reference: Yes) | 1.29 | (0.78–2.12) | 0.319 | 1.29 | (0.78–2.12) | 0.323 | 1.31 | (0.79–2.16) | 0.298 |
| Loneliness | lonely (reference: Not lonely) | 1.62 | (0.96–2.72) | 0.072 | 1.62 | (0.96–2.73) | 0.073 | 1.71 | (1.002–2.92) | 0.049 |
| Subjective Well-Being | 0–11 | 0.87 | (0.82–0.92) | <0.001 | 0.87 | (0.82–0.92) | <0.001 | 0.86 | (0.81–0.92) | <0.001 |

Multiple logistic regression analysis (Forced entry method).

Model 1: Independent variables that were significantly different in univariate analysis were used as covariates.

Significance level of covariates to be entered: <0.1.

Model 2: Control for gender, household composition, marital status, spouse status (bereaved or separated).

Model 3: Control for gender, household composition, marital status, spouse status (bereaved or separated), subjective symptoms of dementia, social isolation.

OR: Odds ratio, CI: Confidence intervals.

(cooking, shopping, money management, etc.), a 3.79 times decline in daily living activities (walking, eating, clothes changing, toilet use, etc.), and a 3.24 times higher depression in their 29-year longitudinal study [38]. Conversely, no association was identified between a sense of economic insecurity and subjective symptoms of dementia, and physical and social conditions, such as the presence or absence of illness, decline in cognitive functioning, and the use of long-term care insurance. Kuiper et al. reported that low social participation was associated with the development of dementia [39]. The frequency of participation in social activities did not differ by age group, with 83.1% of older adults who expressed economic insecurity participating in social activities at least once a week. This result may reflect the characteristics of participants who exercised and interacted with neighbors, as the study was limited to participants who participated in community-based social activities. Comparing the 155 persons excluded from the analysis with those in the analysis as shown in S2 Table, there were no differences in gender, household status, marital status, presence of children, presence of illness, and economic insecurity. However, the mean age of the older adults who were not included in the analysis was 80.9 (SD 6.47), higher than that of the analyzed participants (<0.001), whose the mean age was 78.7 (SD 6.03). Further, the percentage of older adults not included in the analysis with subjective dementia symptoms was 19.3%, higher than the 9.5% of the analyzed participants (<0.001). Additionally, the percentage of older adults not included in the analysis who used long-term care insurance services was 30.5%, higher than the 10.7% of analyzed participants (<0.001). The study population is considered to be a relatively physically and cognitively independent group capable of participating in social activities. The 155 excluded individuals were willing to participate, but were excluded from the analysis because they did not respond to certain items, such as missing 5% or more on each scale or items regarding participation in social activities. They may have found it difficult to answer owing to some influence on their physical, mental, or cognitive functions. This shows that people with dementia symptoms and older adults using long-term care insurance also participate in social activities at the community level. We believe that it is important for older adults to participate in social activities that provide them with opportunities to access a variety of information and to meet kindred spirits in order to prevent the progression of frailty. In future studies, it is necessary to obtain the cooperation of the community general support center staff to expand the number of participants so that all those who participate in social activities can be analyzed.

Nearly half (43.6%) of the older adults aged 65 and older who participate in community activities, expressed a sense of economic insecurity. One possible reason for this may be the prolonged COVID-19 pandemic. Tull et al. reported increased health anxiety, financial worry, and loneliness under a stay-at-home due to the COVID-19 pandemic [40]. The period from July to December 2022, when we conducted our study, was a period of increased infections due to the spread of BA.5, one of the more infectious Omicron strains. Many social activities were carried out after ensuring infection prevention measures. Participation in social activities was voluntary, and it is possible that some older adults refrained from participating. Meanwhile, there were no gender differences in economic insecurity, but 82.6% of participants were women. Since women are more vulnerable to the social and economic effects of the pandemic and an increase in suicides has been reported [41], we believe it is necessary to continuously assess the actual situation of economic insecurity and provide social support to alleviate the sense of insecurity.

The percentage of older adults with economic insecurity was highest among those aged 75–84 years and 0.53 times lower among those aged 85 and older than among those aged 65–74 years in Model 3. As background for the inclusion of economic issues among the anxieties of those in their 60s and 70s, Chatfield compared life satisfaction between the employed and unemployed, and reported that the reason underlying the low life satisfaction among retired

people is the sharp drop in income rather than job loss [42]. The sharp decline in income, longer period between retirement and death, and longer period of time required for nursing care were inferred to have led to a sense of economic insecurity. Therefore, social support may be especially important for retired for older adults between the ages of 65 and 75 years.

In particular, considering that their level of loneliness was 1.71 times higher, the need for interactions with acquaintances and neighbors became particularly important for older adults living alone, as well as for those living with family members. Similar to the results of the present study, in a survey of adults, Rohde et al. reported that economic insecurity affects mental health [43]. Kahn and Pearlin stated that even after excluding the impact of economic situation in old age, long-term economic problems affect physical functioning and depressive symptoms [44]. Cacioppo et al. stated that, in light of the possible role played by loneliness in depressive symptoms, greater attention to loneliness may be important to maximize the likelihood of the maintenance of health and functionality across the life span [45]. A study examining the contribution of various factors to health outcomes found socioeconomic factors to be 47%, health behaviors 34%, and clinical care 16%, indicating the importance of socioeconomic factors [46]. In this study, psychological conditions were associated with a sense of economic insecurity among older adults, highlighting the importance of preventing of early frailty progression through social activities. Gobbens proposed the integral conceptual model of frailty where physical, psychological, and social frailties lead to adverse health outcomes while mutually influencing one another [47]. Poor social participation due to social frailty has negative physical and mental consequences, which in turn lead to a negative cycle of further difficulties in social participation [48]. Maintaining interactions within the community, even in old age, may prevent loneliness and improve subjective health. Therefore, it is important to provide older adults with places where they can gather, participate in social activities at a low cost, and be supported by the staff at the community general support center.

## Implication

Regarding the implications of this study, our results may lead health care providers who promote community-based social participation to consider support tailored to older adults' age and psychological and economic insecurities. Furthermore, given that nearly half of the older adults participating in social activities felt economic insecurity, receiving support to alleviate this issue through social activities could contribute to improving their quality of life, including their mental health.

## Research limitations

This study had several limitations. First, as this was a partial sampling survey conducted during social activities in an urban area of the city with a population of 1.1 million people in Japan, regional bias was not considered. Therefore, the study needs to be expanded in the future and conducted in rural areas. Second, this study measured the subjective economic insecurity of older adults, and the association between actual income and expenses could not be surveyed. Given the cross-sectional nature of the study design, further investigation through longitudinal and intervention studies involving factors based on objective indicators is needed to validate this association. Third, this survey does not reveal the actual situation of older adults in need of care who may have difficulty responding to such a self-administered questionnaire survey. Furthermore, the generalizability of the findings is limited since the study mainly focused on a group of older adults who were independent enough to complete the survey and participate in social activities. The background factors related to older adults' insecurity regarding these issues should be studied further.

## Conclusion

The study showed that 43.6% of older adults who participate in social activities had economic insecurity, and that economic insecurity differed by age group. In particular, the sense of economic insecurity was lower among older adults aged 85 and older, and higher among those who felt psychologically lonely and had a lower subjective sense of well-being. In the future, background factors related to loneliness and subjective well-being should be analyzed, including the number of interactions and the level of intimacy involved in social support.

## Supporting information

**S1 Table. Data for analysis.**
(XLSX)

**S2 Table. Comparison between analyzed and excluded participants.**
(PDF)

## Acknowledgments

The authors would like to thank all staff members of the community general support center who cooperated in this study. The authors would like to thank everyone who participated in the survey. The authors would like to thank Editage (www.editage.com) for English language editing. In writing this paper, the authors would like to thank Dr. Hirofumi Wakaki and Dr. Shinpei Imori of Hiroshima University for lending their expertise on statistical data analysis.

## Author Contributions

**Conceptualization:** Yuriko Inoue.

**Data curation:** Yuriko Inoue, Ichie Ono, Xuxin Peng.

**Formal analysis:** Yuriko Inoue, Xuxin Peng.

**Funding acquisition:** Yuriko Inoue.

**Investigation:** Hisae Nakatani, Ichie Ono, Xuxin Peng.

**Project administration:** Hisae Nakatani.

**Supervision:** Hisae Nakatani.

**Writing – original draft:** Yuriko Inoue.

**Writing – review & editing:** Yuriko Inoue, Hisae Nakatani, Ichie Ono, Xuxin Peng.

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
