## [Decision Letter · Decision Letter 0]

4 Oct 2023

PONE-D-23-12908Factors related to a sense of economic insecurity among older adults who participate in social activitiesPLOS ONE

Dear Dr. Yuriko,

Thank you for submitting your manuscript to PLOS ONE. After careful consideration, we feel that it has merit but does not fully meet PLOS ONE’s publication criteria as it currently stands. Therefore, we invite you to submit a revised version of the manuscript that addresses the points raised during the review process.

We look forward to receiving your revised manuscript.

Kind regards,

Sameh Eltaybani, Ph.D

Academic Editor

PLOS ONE

Journal Requirements:

Additional Editor Comments (if provided):

Dear Authors,

Thank you for your submission.

The Reviewers raised important points that need to be considered before considering the current submission for publication.

Reviewers' comments:

Reviewer's Responses to Questions

**Comments to the Author**

1. Is the manuscript technically sound, and do the data support the conclusions?

Reviewer #1: Partly

Reviewer #2: Partly

2. Has the statistical analysis been performed appropriately and rigorously? 

Reviewer #1: No

Reviewer #2: Yes

3. Have the authors made all data underlying the findings in their manuscript fully available?

Reviewer #1: No

Reviewer #2: No

4. Is the manuscript presented in an intelligible fashion and written in standard English?

Reviewer #1: Yes

Reviewer #2: Yes

5. Review Comments to the Author

Reviewer #1: Dear Authors,

Thank you for your submission

The following may help improve the quality of your report.

1. The Abstract:

Delete the following sentence: This may be because the ....

Line 49: prevents → may prevent

2. INTRODUCTION: Frailty is a state of increased ... Although the Abstract did not mention any thinf about fraility, the first paragraph of the Introduction maily focuses on fraility. So, it is confusing whether the current manuscript targets only frail older adults. You may consider revising the Abstract OR revising the Introduction so that readrs will not get confused.

3. Line 78: The top three causes and motives of suicide → Please describe the three causes and motives in detail.

4. Participants: who participated in social activities → this is a bit unclear. Please explain what social activities are you refering to and why you considered this group of older people to participate in the study. This should be also explained in the Introduction section so that readers would better understand the context of the data collection and the importance of the current studsy to this specific group of older adults.

5. The survey was conducted from July to December 2022. → This means that the survey was conducted during the COVID-19 pandemic, where several social activities and group gathering were prohibitied in may centers in Japan. This might have affected the available population, the response rate, and the results. Please elaborate on these issues in the Discussion.

6. Line 122: Please delete this title (you already have a title for eeach category)

7. Please elaborate on the psychometric properties of all used instruments. For example, did you examine its reliability of the outcome variable? If no, you need to explain why and how the results can be trusted in the absence of psychometeric properties. Also, you need to elaborate on the available tools to assess economic insecurities in the literature.

8. Line 151: UCLA: Mention the full name

9. Excludimng 155 participants from the analysis may introduce bias. Did you consider attrition analysis? Did you consider missing data analysis (such as, MCAR test)? Why did you condier missing data replacement?

10. Please consider draeing a flow chart that show the number of participants at each stage 8recruitment, dropout, etc.)

11. What about sample size calculation?

12. Table 2: It is unclear why you stratified the sample by age group. Please explain in the Method section.

13. Table 2: Please compute the effect size (p-value is not enough)

14. Table 3: Please compute the effect size (p-value is not enough)

15. Data reported in Table 1 ius already reported again in Tables 2 and 3 (pleasee delete any duplication and consider combining tables together).

16. Only 10% of the participants use LTC insurance. Does this mean that the overall condition of the study participants is relatively good, which may affect their response? I am curious to what extentthe current sample reporesnt the community-dwelling older adults in Hiroshima (or in japan). This might hinder the generalizability of the current results.

17. Table 3: the column "Category" is unclear. For example, in "Divorce," are ypou reporting data for "yes" or "no"?

18. Table 3: Did you examine multicollinearity?

19. In the Discussion section, please do not re-write the resuylts again (see for example Line 304).

20. Add a sub-section for the Implication.

===Good Luck===

Reviewer #2: The study investigates the factors associated with the sense of economic insecurity in a select group of older people consisting predominantly of females who participate in government sponsored social activities. The main findings were that the sense of economic insecurity was associated with the age cohort 75-84 years compared with > 84 or 66-74years, increasing loneliness and with subjective sense of well being (inverse association).

Although the overall study design and analysis are appropriate, the authors may wish to address the following:

1) The generalisability of the findings of this study is limited as the sample was derived form a select group that attends social activities. And I believe that this should be adequately emphasised as one of the major limitations.

2) To my understanding , although the authors have explained the rationale for selecting covariates in model 1, they have not done so with respect to models 2 and 3, and some of these variables have collinearity, for example loneliness, social isolation and marital status – perhaps authors may want to comment on this.

3) Some sections under results seem redundant , for example the authors have provided a fairly detailed account of the baseline sample characteristics in tables (table2) and text. I’m not sure such details on sample characteristics are necessary as they are not relevant to the objectives of this study.

4) In the introductory section, the authors state “ For low-cost social activities to function as a place for preventing frailty and suicide, it is necessary to investigate the actual state of the participants’ economic insecurity and discuss how to provide support for effective social activities”(lines 86-88) – are there any references or data to back up this statement.

5) The authors state that the findings of this study are somewhat different from previous studies in that no association was found between physical, cognitive or social aspects and economic insecurity. However, I don’t think that the authors have discussed the likely reasons for these observed differences.

6) I believe that the authors have to be circumspect with respect to some of assertions they have made in the discussion. For example, the authors state ( lines 329-334 “ in this study psychological conditions were associated with a sense of economic insecurity among older adults , highlighting the importance of preventing early frailty progression through social activities. Although the study findings support the 1st part of this statement, viz, in this study psychological conditions were associated with a sense of economic insecurity among older adults, their comments about preventing frailty cannot be inferred from the findings. Similarly, their basis for stating “ to prevent loneliness and improve subjective well being, establishing places where older adults-----------------social activities at low cost will improve their quality of life is not supported by the findings of this study.

6. PLOS authors have the option to publish the peer review history of their article (what does this mean?). If published, this will include your full peer review and any attached files.

Reviewer #1: No

Reviewer #2: No

---

## [Author Response · Author response to Decision Letter 0]

14 Dec 2023

Dear. Reviewer #1

 Thank you for giving us the opportunity to strengthen our manuscript with your valuable comments and queries.

1. The Abstract:

Delete the following sentence: This may be because the ....

Line 49: prevents → may prevent

Reply:

Thank you for your comment. We agree with you and have incorporated these suggestions (p. 2, lines 47–49). 

2. INTRODUCTION: Frailty is a state of increased ... Although the Abstract did not mention any thing about frailty, the first paragraph of the Introduction mainly focuses on frailty. So, it is confusing whether the current manuscript targets only frail older adults. You may consider revising the Abstract OR revising the Introduction so that readers will not get confused.

　

Reply:

Thank you for your comment. We agree with you and have accordingly revised our paper. 

Specifically, we have recast our expression in the Abstract (p. 2, lines 27–29) and Introduction (p. 3–4, lines 59–60,75–79,90–93) to clarify that the paper does not focus only on frail older adults. 

3. Line 78: The top three causes and motives of suicide → Please describe the three causes and motives in detail.

Reply:

Thank you for this suggestion. In response, we have detailed the three causes and motives of suicide. The most common cause and motive of suicide among older adults is "health problems," followed by "family problems," and "economic and lifestyle problems (p. 3, lines 65–69).　

4. Participants: who participated in social activities → this is a bit unclear. Please explain what social activities are you referring to and why you considered this group of older people to participate in the study. This should be also explained in the Introduction section so that readers would better understand the context of the data collection and the importance of the current study to this specific group of older adults.

Reply:

Thank you for your comment. The participants were older adults who participated in social activities that contributed to the prevention of long-term care, such as community-based exercise and hobby activities. The Ministry of Health, Labor and Welfare is working to expand these activities. We have added to and revised the Introduction (p. 4, lines 75–79) and Research Participants (p. 5, lines 100–103) sections to provide a better understanding of our research subjects and the importance of our research.

5. The survey was conducted from July to December 2022. → This means that the survey was conducted during the COVID-19 pandemic, where several social activities and group gathering were prohibited in may centers in Japan. This might have affected the available population, the response rate, and the results. Please elaborate on these issues in the Discussion.

Reply:

Thank you for your feedback. We agree with your assessment. The period from July to December 2022, when we conducted our study, was a period of increased infections due to the spread of BA.5, which was one of the more infectious Omicron strains. Accordingly, many social activities were carried out after ensuring infection prevention measures. In response to your excellent suggestion, we have explained our consideration of the effects of COVID-19 (p. 27–28, lines 344–356).　

6. Line 122: Please delete this title (you already have a title for each category)

Reply:

　Thank you for your suggestion. We accordingly removed this title.

7. Please elaborate on the psychometric properties of all used instruments. For example, did you examine its reliability 　　　　　　　

of the outcome variable? If no, you need to explain why and how the results can be trusted in the absence of psychometric properties. Also, you need to elaborate on the available tools to assess economic insecurities in the literature.

Reply:

　　　Thank you for this query. We have ensured that this information is provided in the Materials and Methods section. We also describe the instruments we used below:

① The self-administered dementia checklist:

The self-administered dementia checklist was developed as a 10-item and 2-factor instrument. Its factorial validity and internal reliability were confirmed. Cronbach’s α for the overall 10-item scale was 0.908. The concurrent and discriminant validities of the checklist were confirmed.

② The Lubben Social Network Scale (LSNS-6):

The Lubben Social Network Scale (LSNS-6) is used worldwide as a screening tool for social isolation in older adults. The reliability and validity of the Japanese version of the LSNS-6 were confirmed. Cronbach’s α was 0.82, the correlation coefficient for the test-retest was 0.92 (P＜0.001), and the intraclass coefficient for interrater reliability was 0.96 (95% confidence interval [CI] 0.90-0.99).

③ The University of California, Los Angeles Loneliness Scale, version 3 (UCLA-LS3):

The University of California, Los Angeles Loneliness Scale, version 3 (UCLA-LS3), is a revised version of the original version of the UCLA-LS by Russell. This revised scale has been adapted and validated in various subjects across different countries. A 3-item version (SF-3), based on the 20-item multidimensional revised UCLA-LS, was developed by Hughes. The three items on the SF-3 were selected because they showed the highest loading on the first factor of a three-factor model. The alpha coefficient for reliability was 0.72.

④ The Philadelphia Geriatric Center (PGC) Morale Scale

 A model containing 11 of the original the Philadelphia Geriatric Center (PGC) Morale Scale items adequately fit both the American and Japanese data.

Based on your feedback, we examined the reliability of the outcome variable and added the finding to the Results (p. 10, lines 225–228).

Regarding the financial aspect, we considered whether we would ask participants about their current income/savings and expenditures. However, we anticipated low response rates, reasoning that some older adults may be reluctant to answer questions relating to their personal information or may find it difficult to answer such questions. We also considered that using a large number of scales would be physically and mentally taxing for older adults. Therefore, in this study, we asked participants about their subjective sense of anxiety about their economic situation. Specifically, to determine their sense of economic insecurity, we asked them to respond to the question, “Do you have economic anxieties?” using a four-point Likert scale (not worried, not too worried, a little worried, worried) (p. 6, lines 123–132).

8. Line 151: UCLA: Mention the full name

Reply:

Thank you for your comment. We added the full name for UCLA (The University of California, Los Angeles) (p. 7, lines 160–161).

9. Excluding 155 participants from the analysis may introduce bias. Did you consider attrition analysis? Did you consider missing data analysis (such as, MCAR test)? Why did you consider missing data replacement?

Reply:

Thank you for these insights. We compared and analyzed the 155 participants who were excluded from the analysis with those who were included in the analysis (Supporting information, Table. Comparison between Analyzed and Excluded Participants). These results were added to the Discussion (p. 27, lines 330–343) and Research limitations (p. 29–30, lines 400–404) sections.

Because the data in this study were not normally distributed, missing data analyses such as MCAR were not performed. 

Among all participants included in the study, social activity participation was unknown for 64 participants and missing answers for basic attributes (e.g., gender, age) and illness were observed for 46 participants; these data were excluded from the analysis. In addition, 45 participants with a deficit of 5% or more in each scale were excluded from the study because of bias introduced by assigning mean values, etc. (Reference 30). (p. 8 lines 173–176). 

10. Please consider drawing a flow chart that show the number of participants at each stage recruitment, dropout, etc.)

Reply:

　　　Thank you for this suggestion. We accordingly developed a flowchart of the participants in the study (Figure 1).

11. What about sample size calculation?

　

Reply:

Thank you for this question. The sample size for this study was analyzed using G*Power. Based on statistical power calculations, the sample size was 602. We added this information to the Analysis Methods section and cited relevant articles (p. 5, lines 102–106).

12. Table 2: It is unclear why you stratified the sample by age group. Please explain in the Method section.

Reply:

Thank you for this query. Life expectancy is increasing globally, and the number of older adults aged 85 and over is accordingly growing. In response to your excellent suggestion, we added our rationale for stratifying our sample by age group to the Analysis Methods section; this decision was based on prior research (Reference 31) (p. 8, lines 176–181).

13. Table 2: Please compute the effect size (p-value is not enough)

Reply:

Thank you for this suggestion. We computed the effect sizes and included them in Table 2 (p. 12–16, lines 254–259).

14. Table 3: Please compute the effect size (p-value is not enough)

Reply:

Thank you for this guidance. As above, we computed the effect sizes and included them in Table 3 (p. 18–22, lines 273–277).

15. Data reported in Table 1 is already reported again in Tables 2 and 3 (please delete any duplication and consider combining tables together).

　

Reply:

Thank you for pointing this out. We combined the tables and removed the table that was originally Table 1.

16. Only 10% of the participants use LTC insurance. Does this mean that the overall condition of the study participants 　　　　　

is relatively good, which may affect their response? I am curious to what extent the current sample represent the community-dwelling older adults in Hiroshima (or in Japan). This might hinder the generalizability of the current results.

Reply:

Thank you for this question. The older adults in the study were relatively physically and cognitively independent and able to complete the self-administered questionnaire survey and participate in social activities. According to a report on the implementation of the long-term care insurance system in Hiroshima Prefecture, in 2022, the percentage of people certified as having a low long-term care insurance level (support needs: levels 1 and 2, and care needs level 1) in Japan was 9.3% (10.1% in Hiroshima Prefecture), which was similar to the 10.7% rate in this study. We added this information to the Discussion (p. 26, lines 308–316) and included more References to substantiate these claims (Reference 34–36). 

17. Table 3: the column "Category" is unclear. For example, in "Divorce," are you reporting data for "yes" or "no"?

　

Reply:

Thank you for checking in about this. We have revised Table 3 to clarify this point (we are reporting data for yes/no responses). Additionally, we modified our results to more closely report the results of the logistic analysis by age group (p. 23–25, lines 280–304).

18. Table 3: Did you examine multicollinearity?

　

Reply:

Thank you for this query. We added a discussion of multicollinearity to the Results (p. 23, lines 283–284; 

p. 23, 288–289 and 295–296).

19. In the Discussion section, please do not re-write the results again (see for example Line 304).

Reply:

Thank you for this insight. We agree and have incorporated this suggestion in our revised manuscript. 

20. Add a sub-section for the Implication.

Reply:

Thank you very much for this guidance. We accordingly added an Implication subsection (p. 29, lines 388–392). The implication of this study is that health professionals who promote community-based social participation can provide support tailored to older adults' specific ages and levels of psychological and economic insecurity. This can improve older adults’ quality of life, including their mental health, and prevent frailty.

Dear. Reviewer #2

 Thank you for giving us the opportunity to strengthen our manuscript with your valuable comments and queries.

The study investigates the factors associated with the sense of economic insecurity in a select group of older people consisting predominantly of females who participate in government sponsored social activities. The main findings were that the sense of economic insecurity was associated with the age cohort 75-84 years compared with > 84 or 66-74years, increasing loneliness and with subjective sense of well-being (inverse association).

Although the overall study design and analysis are appropriate, the authors may wish to address the following:

1) The generalizability of the findings of this study is limited as the sample was derived form a select group that attends social activities. And I believe that this should be adequately emphasized as one of the major limitations.

Reply:

 Thank you very much for your excellent suggestion. We agree. As you pointed out, the study population was rather specific; it comprised older adults who were relatively physically and cognitively independent and who were able to complete a self-administered questionnaire survey and participate in social activities. In response, we highlighted the limited generalizability of the findings in the Discussion (p. 26, lines 308–316; p. 27, lines 330–343) and Limitations (p. 29–30, lines 400–404).

2) To my understanding, although the authors have explained the rationale for selecting covariates in model 1, they have not done so with respect to models 2 and 3, and some of these variables have collinearity, for example loneliness, social isolation and marital status – perhaps authors may want to comment on this.

Reply:

Thank you for your suggestion. We added our rationale for the selection of the covariates in Models 2 and 3 and detailed the multicollinearity in the Results (p. 23, lines 284–296). 

3) Some sections under results seem redundant, for example the authors have provided a fairly detailed account of the baseline sample characteristics in tables (table2) and text. I’m not sure such details on sample characteristics are necessary as they are not relevant to the objectives of this study.

Reply:

 Thank you for these insights. Life expectancy is increasing globally, and the number of older adults aged 85 and over is accordingly growing. In response to your excellent suggestion, we added our rationale for stratifying our sample by age group to the Analysis Methods section; this decision was based on prior research (Reference 31) (p. 8, lines 176–181). To remove the redundancies in the results section, we have combined the tables and removed what was originally Table 1.

4) In the introductory section, the authors state “For low-cost social activities to function as a place for preventing frailty and suicide, it is necessary to investigate the actual state of the participants’ economic insecurity and discuss how to provide support for effective social activities” (lines 86–88) are there any references or data to back up this statement.

Reply:

 Thank you for checking in about this. Prior research reported that community-level social participation and support prevent suicide (Reference 15). De Leo reported that suicide prevention for old adults should focus on the number of socio-environmental conditions that can be particularly worrisome in old age, such as decreased physical health, social isolation and loneliness, and economic insecurity (Reference 16). We added new text and references to the Introduction to substantiate these claims (p. 4, lines 82–86).

5) The authors state that the findings of this study are somewhat different from previous studies in that no association was found between physical, cognitive or social aspects and economic insecurity. However, I don’t think that the authors have discussed the likely reasons for these observed differences.

Reply:

 Thank you very much for this observation. In response, we included some possible reasons for these differences in the Discussion section (Reference 38) (p. 26–27, lines 324–343).

6) I believe that the authors have to be circumspect with respect to some of assertions they have made in the discussion. For example, the authors state (lines 329–334 “in this study psychological conditions were associated with a sense of economic insecurity among older adults, highlighting the importance of preventing early frailty progression through social activities. Although the study findings support the 1st part of this statement, viz, in this study psychological conditions were associated with a sense of economic insecurity among older adults, their comments about preventing frailty cannot be inferred from the findings. Similarly, their basis for stating “to prevent loneliness and improve subjective well-being, establishing places where older adults-----------------social activities at low cost will improve their quality of life is not supported by the findings of this study.

Reply:

Thank you for this feedback. In this study showed that psychological status was associated with economic insecurity among older adults. As your comment, we cannot infer from the results of this study regarding the prevention of frailty. We agree with your assessment and have made according changes to the Discussion (p. 29, lines 384–387) and the Abstract (p. 2, lines 47–49). Notably, we added a reference (Reference 45-47) in our revisions based on your comment to the Discussion. We also highlighted that a key implication of this study is that social participation can prevent frailty (p. 29, lines 388–392).

---

## [Decision Letter · Decision Letter 1]

16 Jan 2024

PONE-D-23-12908R1Factors related to a sense of economic insecurity among older adults who participate in social activitiesPLOS ONE

Dear Dr. Yuriko,

Thank you for submitting your manuscript to PLOS ONE. After careful consideration, we feel that it has merit but does not fully meet PLOS ONE’s publication criteria as it currently stands. Therefore, we invite you to submit a revised version of the manuscript that addresses the points raised during the review process.

We look forward to receiving your revised manuscript.

Kind regards,

Sameh Eltaybani, Ph.D

The University of Tokyo

Academic Editor

PLOS ONE

Journal Requirements:

Additional Editor Comments:

Thank you for the thorough revision. The current version of the manuscript looks far better than the previous one. Yet, the Reviewer still pointed out some issues that need to be considered before considering the current manuscript for publication.

Reviewers' comments:

Reviewer's Responses to Questions

**Comments to the Author**

1. If the authors have adequately addressed your comments raised in a previous round of review and you feel that this manuscript is now acceptable for publication, you may indicate that here to bypass the “Comments to the Author” section, enter your conflict of interest statement in the “Confidential to Editor” section, and submit your "Accept" recommendation.

Reviewer #1: All comments have been addressed

Reviewer #2: All comments have been addressed

2. Is the manuscript technically sound, and do the data support the conclusions?

Reviewer #1: (No Response)

Reviewer #2: Partly

3. Has the statistical analysis been performed appropriately and rigorously? 

Reviewer #1: (No Response)

Reviewer #2: Yes

4. Have the authors made all data underlying the findings in their manuscript fully available?

Reviewer #1: (No Response)

Reviewer #2: No

5. Is the manuscript presented in an intelligible fashion and written in standard English?

Reviewer #1: (No Response)

Reviewer #2: Yes

6. Review Comments to the Author

Reviewer #1: (No Response)

Reviewer #2: 1) The references 15 and 16 in the introduction, the authors have now cited in their response, refer to the association between environmental factors /activities and suicide or suicide prevention, and are not relevant frailty. Therefore, they are not the appropriate references for suggesting any potential association between the sense of economic insecurity or social activities and frailty.

2) The authors now mention the sample size but fail to discuss estimated power or set alpha values.

3) The authors may wish to consider sensitivity analyses including the 155 participants excluded from analysis, as there are significant differences in their baseline characteristics - they are significantly older, more likely to have dementia symptoms and use long-term care insurance.

4) Under implications, the authors state “ Regarding the implication of this study, it reveals that health professionals who promote community- based social participation can provide support tailored to older adults’ age and level of psychological and economic insecurity . This work can improve older adults’ quality of life , including their mental health, and prevent frailty”. I’m not sure one could draw these conclusions as they are not underpinned by the findings of this study – the study did not investigate the potential consequences or implications of the sense of economic security, or the possible interventions to ameliorate the sense of economic security. Similarly, the concluding section too was outside the scope of the findings of this study. For example, I don’t believe that the findings of this study support the following conclusion – “Notably , this study showed that preventive measures to limit loneliness and increase subjective well-being through the use of social activities play an important role in reducing economic insecurity”.

7. PLOS authors have the option to publish the peer review history of their article (what does this mean?). If published, this will include your full peer review and any attached files.

Reviewer #1: No

Reviewer #2: No

---

## [Author Response · Author response to Decision Letter 1]

2 Feb 2024

Dear. Reviewer #2

Thank you for giving us the opportunity to strengthen our manuscript with your valuable comments and queries.

1) The references 15 and 16 in the introduction, the authors have now cited in their response, refer to the association between environmental factors /activities and suicide or suicide prevention, and are not relevant frailty. Therefore, they are not the appropriate references for suggesting any potential association between the sense of economic insecurity or social activities and frailty.

　

Reply:

Thank you for pointing this out. In response to your suggestion, we removed Reference 15 because it was an article that referred to the association between environmental factors/activities and suicide or suicide prevention. Reference 16 has been moved to the section related to suicide in the Introduction (accordingly, Reference 16 is Reference 9 in the revised manuscript) (p. 3, lines 69–72). Additionally, we have added references that demonstrate any potential association between the sense of economic insecurity or social activities and frailty (References 16 and 17) (p. 4, lines 86–91). 

2) The authors now mention the sample size but fail to discuss estimated power or set alpha values.

　

Reply:

Thank you for this suggestion. We set a power of 0.80 and an alpha value of 0.05. We have added this information to the Research participants subsection of the Methods section (p. 5, line 108).

3) The authors may wish to consider sensitivity analyses including the 155 participants excluded from analysis, as there are significant differences in their baseline characteristics - they are significantly older, more likely to have dementia symptoms and use long-term care insurance.

Reply:

Thank you for your feedback. The 155 excluded individuals were willing to participate, but were excluded from the analysis because they did not respond to certain items, such as missing 5% or more on each scale or items regarding participation in social activities. They may have found it difficult to answer owing to some influence on their physical, mental, or cognitive functions. This shows that people with dementia symptoms and older adults using long-term care insurance also participate in social activities at the community level. We believe that it is important for older adults to participate in social activities that provide them with opportunities to access a variety of information and to meet kindred spirits in order to prevent the progression of frailty. In future studies, it is necessary to obtain the cooperation of the community general support center staff to expand the number of participants so that all those who participate in social activities can be analyzed. We have added the underlined section to the Discussion (p. 27, lines 345–355).

4) Under implications, the authors state “ Regarding the implication of this study, it reveals that health professionals who promote community- based social participation can provide support tailored to older adults’ age and level of psychological and economic insecurity . This work can improve older adults’ quality of life , including their mental health, and prevent frailty”. I’m not sure one could draw these conclusions as they are not underpinned by the findings of this study – the study did not investigate the potential consequences or implications of the sense of economic security, or the possible interventions to ameliorate the sense of economic security. Similarly, the concluding section too was outside the scope of the findings of this study. For example, I don’t believe that the findings of this study support the following conclusion – “Notably, this study showed that preventive measures to limit loneliness and increase subjective well-being through the use of social activities play an important role in reducing economic insecurity”.

Reply:

Thank you for these insights. We agree. As you pointed out, the design of this study is cross-sectional, and further longitudinal and intervention studies are needed in the future to verify the associations. In response, we have revised and deleted the parts of the Implication (p. 29, lines 402–404), Limitations (p. 30, lines 412–414), and Conclusions (“Notably, this study showed that preventive measures to limit loneliness and increase subjective well-being through the use of social activities play an important role in reducing economic insecurity”.) that were pointed out by you.

---

## [Decision Letter · Decision Letter 2]

14 Feb 2024

PONE-D-23-12908R2Factors related to a sense of economic insecurity among older adults who participate in social activitiesPLOS ONE

Dear Dr. Yuriko,

Thank you for submitting your manuscript to PLOS ONE. After careful consideration, we feel that it has merit but does not fully meet PLOS ONE’s publication criteria as it currently stands. Therefore, we invite you to submit a revised version of the manuscript that addresses the points raised during the review process.

We look forward to receiving your revised manuscript.

Kind regards,

Sameh Eltaybani, Ph.D

The University of Tokyo

Academic Editor

PLOS ONE

Journal Requirements:

**Additional Editor Comments:**

Thank you for the thorough revision. The quality of the current version is by far much better than the previous version. Yet, some minor issues need to be addressed.

1) The sample size calculation is unclear. The mentioned description is insufficient. You need to describe in detail how the sample size was computed including the name of the software used.

2) In the STATISTICAL ANALYSIS section, p-value is on-tailed or two tailed?

3) Table 1: you used symbols a,b and A,B,C. This is confusing. Do not use capital and small forms of the same letter to denote different things. Please use different symbols. Also, reporting the results of the post-hoc analysis (A,B,C) is unclear. You may need to consult a statistician about how to report the results of the post-hoc analysis clearly. For example, what is the meaning of BC and ABC?

4) You need to elaborate on the IMPLICATIONS of the current study (Line 401).

5) Figure 1: The quality is extremely low. Please provide a hig-quality figure.

Reviewers' comments:

Reviewer's Responses to Questions

**Comments to the Author**

1. If the authors have adequately addressed your comments raised in a previous round of review and you feel that this manuscript is now acceptable for publication, you may indicate that here to bypass the “Comments to the Author” section, enter your conflict of interest statement in the “Confidential to Editor” section, and submit your "Accept" recommendation.

Reviewer #1: All comments have been addressed

Reviewer #2: All comments have been addressed

2. Is the manuscript technically sound, and do the data support the conclusions?

Reviewer #1: (No Response)

Reviewer #2: Yes

3. Has the statistical analysis been performed appropriately and rigorously? 

Reviewer #1: (No Response)

Reviewer #2: Yes

4. Have the authors made all data underlying the findings in their manuscript fully available?

Reviewer #1: (No Response)

Reviewer #2: (No Response)

5. Is the manuscript presented in an intelligible fashion and written in standard English?

Reviewer #1: (No Response)

Reviewer #2: Yes

6. Review Comments to the Author

Reviewer #1: (No Response)

Reviewer #2: (No Response)

7. PLOS authors have the option to publish the peer review history of their article (what does this mean?). If published, this will include your full peer review and any attached files.

Reviewer #1: No

Reviewer #2: No

---

## [Author Response · Author response to Decision Letter 2]

25 Feb 2024

Responses to Reviewers’ Comments

Manuscript Number：PONE-D-23-12908R2

Full Title：Factors related to a sense of economic insecurity among older adults who participate in social activities

Dear. Editor

 Thank you for giving us the opportunity to revise and strengthen our manuscript based on your valuable comments and queries.

1) The sample size calculation is unclear. The mentioned description is insufficient. You need to describe in detail how the sample size was computed including the name of the software used.

Reply:

Thank you for your comment. We used G*Power to calculate our sample size. The Cabinet Office 2020 White Paper on Older Adults reports that 74.1%, or three-quarters of those aged 60 and older, are living without economic insecurity, more than that in the 2016 survey (see Annual Report on the Ageing Society [Summary] FY2020: https://www8.cao.go.jp/kourei/english/annualreport/2020/pdf/2020.pdf). However, we expected the percentage of older adults with economic insecurity to increase to approximately 30% with the COVID-19 pandemic. We predicted that those with good physical, social, and psychological status would be 0.8 times more likely to have economic insecurity than those with low economic insecurity, which we set at 602 on a one-tailed test. We also examined the sample size based on the Events per variable following Peduzzi et al., who reported that data should be collected so that the Events per variable is greater than 10 (Peduzzi et.al., 1996). We examined the following explanatory variables to identify the factors associated with economic insecurity: basic attributes: 1) gender, 2) age, 3) household composition, 4) marital status, spouse status (5) bereaved, 6) divorced, 7) separated), 8) whether the participant had a child (children); physical conditions:9) presence of illness, 10) subjective symptoms of dementia, 11) whether long-term care insurance services were used; social conditions:12) frequency of outings, 13) frequency of participation in social activities, 14) working status, 15) the presence or absence of social isolation; and psychological conditions: 16) loneliness, and 17) subjective well-being. Assuming 30% of the respondents have a sense of economic insecurity, the sample size must be at least 567.

2) In the STATISTICAL ANALYSIS section, p-value is on-tailed or two tailed?

Reply:

Thank you for your concern. We performed a two-tailed test. We have added the following sentence to the manuscript to reflect this: “A two-tailed p-value of <0.05 was considered to indicate statistical significance.” (p. 9, lines 212-213).

3) Table 1: you used symbols a,b and A,B,C. This is confusing. Do not use capital and small forms of the same letter to denote different things. Please use different symbols. Also, reporting the results of the post-hoc analysis (A,B,C) is unclear. You may need to consult a statistician about how to report the results of the post-hoc analysis clearly. For example, what is the meaning of BC and ABC?

Reply:

Thank you for this suggestion. We have revised the symbols “a” to “†” and “b” to “‡”. We have also corrected the reporting of the post-hoc analysis results to “†: χ² and z-tests were performed to compare column proportions. p-values have been Bonferroni corrected. a: vs 65-74 years old, b: vs 75-84 years old, p<0.05” (p. 12-16, lines 259-262).

 References:

 1) Mielgo-Ayuso J, Aparicio-Ugarriza R, Castillo A, Ruiz E, Ávila JM, Aranceta-Batrina J, et al. Physical activity patterns of the Spanish population are mostly determined by sex and age: Findings in the ANIBES study. PLoS One. 2016;11(2):e0149969. doi: 10.1371/journal.pone.0149969.

 2) Koyama A, Hashimoto M, Tanaka H, Fujise N, Matsushita M, Miyagawa Y, et al. Malnutrition in Alzheimer’s disease, dementia with Lewy bodies, and frontotemporal lobar degeneration: Comparison using serum albumin, total protein, and hemoglobin level. PLoS One. 2016;11(6):e0157053. doi: 10.1371/journal.pone.0157053.

4) You need to elaborate on the IMPLICATIONS of the current study (Line 401). 

Reply:

Thank you very much for this suggestion. The implications were suggested by Reviewer #1, and in the second draft, the following text was added: “Regarding the implication of this study, it reveals that health professionals who promote community-based social participation can provide support tailored to older adults' age and levels of psychological and This work can improve older adults' quality of life, including their mental health, and prevent frailty.” In the third draft, we revised parts of the text in accordance with the suggestions we received from reviewer #2 as follows: “Regarding the implications of this study, our results may lead health care providers who promote community-based social participation to consider support tailored to older adults’ age and psychological and economic insecurities.” We agree with your suggestion to elaborate on the implications, so we have further revised the paper in Draft 4 as follows: “Regarding the implications of this study, our results may lead healthcare providers who promote community-based social participation to consider support tailored to older adults’ age and psychological and economic insecurities. Furthermore, given that nearly half of the older adults participating in social activities felt economic insecurity, receiving support to alleviate this issue through social activities could contribute to improving their quality of life, including their mental health.” (p. 29, lines 401–407).

5) Figure 1: The quality is extremely low. Please provide a hig-quality figure.

Reply:

Following on your suggestion, we have re-created Figure 1 to ensure that it is of high-quality.

---

## [Editor Report · Decision Letter 3]

14 Mar 2024

Factors related to a sense of economic insecurity among older adults who participate in social activities

PONE-D-23-12908R3

Dear Dr. Yuriko,

We’re pleased to inform you that your manuscript has been judged scientifically suitable for publication and will be formally accepted for publication once it meets all outstanding technical requirements.

Kind regards,

Sameh Eltaybani, Ph.D

The University of Tokyo
---

## [Editor Report · Acceptance letter]

19 Mar 2024

PONE-D-23-12908R3 

PLOS ONE

Dear Dr. Inoue, 

I'm pleased to inform you that your manuscript has been deemed suitable for publication in PLOS ONE. Congratulations! Your manuscript is now being handed over to our production team.

Kind regards, 

on behalf of

Dr. Sameh Eltaybani 

Academic Editor

PLOS ONE